# Growth, Enzymatic, and Transcriptomic Analysis of *xyr1* Deletion Reveals a Major Regulator of Plant Biomass-Degrading Enzymes in *Trichoderma harzianum*

**DOI:** 10.3390/biom14020148

**Published:** 2024-01-24

**Authors:** Lunji Wang, Yishen Zhao, Siqiao Chen, Xian Wen, Wilfred Mabeche Anjago, Tianchi Tian, Yajuan Chen, Jinfeng Zhang, Sheng Deng, Min Jiu, Pengxiao Fu, Dongmei Zhou, Irina S. Druzhinina, Lihui Wei, Paul Daly

**Affiliations:** 1College of Food and Bioengineering, Henan University of Science and Technology, Luoyang 471023, China; lunjiwang@haust.edu.cn (L.W.); 210321090599@stu.haust.edu.cn (Y.Z.); 210321090590@stu.haust.edu.cn (X.W.); jiumin0912@haust.edu.cn (M.J.); 2Key Lab of Food Quality and Safety of Jiangsu Province—State Key Laboratory Breeding Base, Institute of Plant Protection, Jiangsu Academy of Agricultural Sciences, Nanjing 210014, China; 33315220@njau.edu.cn (S.C.); 20230010@jaas.ac.cn (W.M.A.); 13861928779@163.com (T.T.); ts22040220p31@cumt.edu.cn (Y.C.); 20140980@jaas.ac.cn (J.Z.); dengsheng@jaas.ac.cn (S.D.); dongmeizhou@jaas.ac.cn (D.Z.); 3Fungal Genomics Laboratory (FungiG), Nanjing Agricultural University, Nanjing 210095, China; 4Key Laboratory of Coal Processing and Efficient Utilization, China University of Mining and Technology, Xuzhou 221116, China; 5Jiangsu Coastal Ecological Science and Technology Development Co., Ltd., Nanjing 210036, China; fpx960610@foxmail.com; 6Department of Accelerated Taxonomy, The Royal Botanic Gardens Kew, London TW9 3AE, UK; i.druzhinina@kew.org

**Keywords:** CAZymes, XYR1/XlnR/XLR-1, cellulose, transcriptional regulation

## Abstract

The regulation of plant biomass degradation by fungi is critical to the carbon cycle, and applications in bioproducts and biocontrol. *Trichoderma harzianum* is an important plant biomass degrader, enzyme producer, and biocontrol agent, but few putative major transcriptional regulators have been deleted in this species. The *T*. *harzianum* ortholog of the transcriptional activator XYR1/XlnR/XLR-1 was deleted, and the mutant strains were analyzed through growth profiling, enzymatic activities, and transcriptomics on cellulose. From plate cultures, the Δ*xyr1* mutant had reduced growth on D-xylose, xylan, and cellulose, and from shake-flask cultures with cellulose, the Δ*xyr1* mutant had ~90% lower β-glucosidase activity, and no detectable β-xylosidase or cellulase activity. The comparison of the transcriptomes from 18 h shake-flask cultures on D-fructose, without a carbon source, and cellulose, showed major effects of XYR1 deletion whereby the Δ*xyr1* mutant on cellulose was transcriptionally most similar to the cultures without a carbon source. The cellulose induced 43 plant biomass-degrading CAZymes including xylanases as well as cellulases, and most of these had massively lower expression in the Δ*xyr1* mutant. The expression of a subset of carbon catabolic enzymes, other transcription factors, and sugar transporters was also lower in the Δ*xyr1* mutant on cellulose. In summary, *T*. *harzianum* XYR1 is the master regulator of cellulases and xylanases, as well as regulating carbon catabolic enzymes.

## 1. Introduction

The *Trichoderma* genus includes important biocontrol agents and plant growth-promoting filamentous fungi, as well as saprotrophic fungi that are critical to nutrient cycles [1]. *Trichoderma* is found in diverse habitats such as soil, water, and the rhizosphere of plants [2]. The efficient degradation of plant biomass by *Trichoderma* species relies on the production of a diverse range of plant biomass-degrading enzymes, such as cellulases and hemicellulases, and requires a gene regulation system that can tightly control the expression of the relevant genes to optimize enzyme production under appropriate conditions [3,4]. *Trichoderma* species are known to be highly efficient producers of plant biomass-degrading enzymes such as cellulases, making them a valuable source for industrial applications such as plant biomass conversion and biofuel production. In particular, cellulase enzymes play a crucial role in the breakdown of cellulose, one of the most abundant polysaccharides on Earth. *Trichoderma harzianum* is one of the most important *Trichoderma* species, and is recognized for its significant potential in various biotechnological applications, such as plant disease biocontrol, plant growth promotion [5], and plant biomass-degrading enzyme production [6].

The transcriptional regulation of plant biomass-degrading enzymes in filamentous fungi is a complex process involving a range of activating and repressing transcription factors that are signaled to via a range of environmental signals, with inducing sugars derived from plant biomass being the most prominent environmental signals [3,4,7]. Within ascomycete fungi, there are lineage-specific transcription factors as well as more taxonomically broad conserved transcription factors, and a further layer of complexity is added by the variation in the range of enzymes activated by orthologous transcription factors. E.g., XYR1/XlnR/XLR-1 is one of the major transcriptional activators of plant biomass-degrading enzymes, and can activate cellulase and xylanase gene expression in fungi such as *Aspergillus niger* [8] and *Trichoderma reesei* [9], and XYR1/XlnR can in effect act as a master regulator. In contrast, in another ascomycete, *Neurospora crassa*, the activation is limited mainly to xylanase gene expression [10]. As well as XYR1/XlnR/XLR-1, other transcriptional activators of (hemi-)cellulose expression include ACE2 [11], ACE3 [12], and CLR-2 [13]. The transcription factor XPP1 has been identified as a xylanase but not a cellulase regulator in *T. reesei* [14]. Other regulators activate a narrower range of enzyme activities, such as the arabinose- and galactose-related ARA1 transcription factor [15]. As well as activators, there can be repressors that can be either wide-domain-acting, such as the carbon catabolite repressor CRE1/CreA [16], or plant biomass-degrading enzyme-specific transcriptional repressors such as ACE1 [17] and RCE1 [18]. As well as transcription factors, the regulation of xylanase and cellulase expression is related to other factors such as how in *T. reesei*, the packaging of chromatin is an important factor [19].

Within the *Trichoderma* genus, there are only three species where *xyr1* has been deleted, and these are *T. reesei* (Section Longibrachiatum), *T*. *atroviride* (Section Trichoderma), and *T*. cf. *guizhouense* (Harzianum/Virens clade). In *T. reesei*, XYR1 is a transcription factor that acts as a master regulator, controlling the expression of cellulases and hemicellulases in response to the presence of cellulose and hemicellulose substrates. In *T*. *reesei* QM9414, the deletion of *xyr1* led to a lower expression of cellulases and xylanases on crude plant biomass substrates [20] and Avicel cellulose [21]. Similarly, in *T*. *reesei* RUT-C30, the deletion of *xyr1* led to a lower expression of most cellulases and hemicellulases, and many sugar transporters, when cultured on a mixture of Avicel cellulose and wheat bran [22]. In the *T*. *reesei* RUT-C30 Δ*xyr1* mutant, other genes had an increased expression related to a suggested starvation stress response due to the inability of the Δ*xyr1* mutant to obtain carbon from the complex plant biomass substrates [22]. As well as regulating the plant biomass-degrading enzymes, *T*. *reesei* XYR1 can also regulate genes involved in the catabolism of D-xylose, as evidenced by the dramatically reduced growth of the *T. reesei* QM9414 Δ*xyr1* mutant on D-xylose [9]. In *T*. *reesei* QM9414, XYR1 is also involved in co-regulation with other transcription factors; e.g., ARA1 and XYR1 co-regulate the transcriptional response to L-arabinose [15]. In the *Trichoderma* cf. *guizhouense* strain NJAU4742, XYR1 appeared to be the main regulator of cellulases and xylanases, although the induction pattern of cellulases and xylanases varied depending on whether Avicel cellulose or xylan was used as the inducer [23]. Although extensive research has elucidated the role of *xyr1* in cellulase and hemicellulase production in *T*. *reesei*, and more recently in *T*. cf. *guizhouense*, there are no reports of the regulon of XYR1 in the *Trichoderma atroviride* P1 strain Δ*xyr1* mutant, as the study focused on other aspects of XYR1 function [24]. There are also no reports of the deletion of *xyr1* in *T. harzianum*.

Previously, *xyr1* was overexpressed in the *T. harzianum* strain P49P11 under the control of a constitutive promoter [25]. In the *T. harzianum xyr1* overexpressing strain, there was ~50% higher cellulase, xylanase, and β-glucosidase activities, and a higher expression of cellulases and xylanases was measured in cultures with sugar cane bagasse [25]. The higher levels of gene expression and activities in the *T. harzianum* P49P11 *xyr1* overexpressing strain suggested that XYR1 in *T. harzianum* could be co-regulating the expression of cellulases and xylanases, but an *xyr1* deletion mutant would be required to investigate this more conclusively. A further recent analysis of a set of wild type *T. harzianum* strains suggested an extensive *xyr1* co-expression gene network, which is potentially regulated by XYR1 [26]. In our study, we deleted *xyr1* in *T. harzianum* and analyzed growth patterns, enzymatic activities, and transcriptional responses to cellulose, to understand the function of *T. harzianum* XYR1.

## 2. Materials and Methods

### 2.1. Strains, Media, and General Growth Conditions

*Escherichia coli* DH5α was utilized for routine cloning, and the strain *Trichoderma harzianum* CBS 226.95 was used as the parental strain for gene knockout. All *T. harzianum* plate cultures were incubated at 28 °C on PDA for sporulation, or minimal medium with 15 g/L agar during the transformation work. The composition of the minimal medium (MM) used was 5 g/L (NH_4_)_2_SO_4_, 15 g/L KH_2_PO_4_, 0.6 g/L MgSO_4_, 0.6 g/L CaCl_2_, and 2 mL/L Vishniac solution 500X stock, and adjusted to pH 5.5 with KH_2_PO_4_ or K_2_HPO_4_ [27].

### 2.2. Construction of xyr1 Deletion T. harzianum Strains

PEG-mediated protoplast transformation of *T. harzianum* was performed based on the protocol described previously [28]. Appendix A lists the primer sequences used in the generation of the deletion fragment, the screening of transformants, and the confirmation of deletion mutants, and Figure 1 illustrates the binding sites of these primers. The hygromycin resistance gene cassette was amplified from the pcDNA1 plasmid, and combined with ~2000 bp upstream and downstream flanks of the *xyr1* CDS (Triha1_1126), and the pCE-Zero vector using the recombination-based ClonExpress Ultra One Step cloning kit (Vazyme, Nanjing, China). The deletion fragment was amplified from deletion plasmids, and 1 µg was used to transform protoplasts, and 120 µg/mL hygromycin was used to select for transformants. After PCR-based screening of transformants, single-spore purification, and culturing on and off hygromycin selection to confirm the stability of strains was performed. The strain spore suspensions were stored at −80 °C in 30% glycerol. Two independent *xyr1* deletion mutants were generated (Δ*xyr1-2* and Δ*xyr1-63*). The gene modifications in the recombinant strains were confirmed using PCR and the sequencing of the PCR products at Sangon Biotech, Shanghai, China.

### 2.3. Growth Profiling Analysis of T. harzianum Δxyr1 Mutants

The growth profiling was performed in triplicate in 9 cm Petri dishes on MM with 1.5% Ultra-pure agarose (Invitrogen, Carlsbad, CA, USA) with one of the following carbon sources: 25 mM D-glucose (Cat. no. G6172, Macklin, Shanghai, China), D-fructose (Cat. no. D809612, Macklin), D-xylose (Cat. no. XBO998, Sangon Biotech, Shanghai, China), L-arabinose (Cat. no. L824031, Macklin), cellobiose (Cat. no. C6182, Macklin), 1% *w*/*v* microcrystalline cellulose from cotton linters (Cat. no. 435236, Sigma, Shanghai, China), or 1% *w*/*v* xylan (from corn cob) (Cat. no. X823251, Macklin), or without an added carbon source. For growth profiling plate cultures, the plates were inoculated with a ~4 mm^2^ agar plug from the growing edge of the *T. harzianum* colony from a PDA plate, and then incubated for at least four days at 28 °C with 12 h light and 12 h dark. After 48 h, the colony diameters were measured, and after both 48 h and 96 h, photos of the colonies were taken. Appendix A contains the colony diameter measurements from the two repeat experiments. Two independent deletion strains were tested for growth on the carbon sources to confirm the reliability of attributing the observed phenotypes to the deleted gene. We selected one of the deletion strains to use in further studies.

### 2.4. Shake-Flask Cultures for Enzymatic and Transcriptomic Analyses

To inoculate pre-cultures, spores were harvested from a PDA plate using a 0.05% tween 20 solution, filtered with a nappy gauze, and inoculated into a 250 mL flask containing 50 mL MM supplemented with 1% *w*/*v* D-fructose and 0.1% *w*/*v* peptone to give a 1 × 10^4^/mL spore final concentration. The pre-cultures were incubated at 28 °C and 200 RPM for 36 h in the dark. For transfer, the mycelia were collected using a 38 µm nylon net filter and washed twice with ~200 mL MM solution. Then, 1 g wet weight of mycelia was transferred into 250 mL flasks containing 50 mL of MM with either 25 mM D-fructose, 1% *w*/*v* cellulose (microcrystalline from cotton linters), or without an added carbon source, and incubated for up to 36 h at 28 °C with 200 RPM in the dark. Liquid samples from the cultures were collected at various time-points for enzyme assays and PAGE gel analysis. The liquid samples from the cultures were centrifuged, and the supernatants were flash-frozen in liquid nitrogen and stored at −20 °C. For transcriptomic analysis, mycelia were collected from cultures from the 18 h time-point, flash-frozen in liquid nitrogen, and stored at −80 °C. To take account of variability from different spore plates, separate spore plates were used to inoculate each of the three sets of replicate pre-cultures, and each set of replicate pre-cultures was transferred to their respective replicate cultures with various carbon sources.

### 2.5. Enzyme Activity and PAGE Gel Analysis of the T. harzianum Δxyr1 Mutant

*p*-nitrophenol (*p*NP) enzymatic activity assays were performed in 96-well plates using a total volume of 100 μL containing 20 mM Tris-HCl pH 7.5, 2.5 mM of the *p*-nitrophenol-linked substrate, and culture supernatants. A range of concentrations of *p*-nitrophenol standard solution (0.5 to 25 nmol/100 µL) were prepared to determine the concentration of *p*-nitrophenol released. The absorbance was measured at 405 nm using a spectrophotometric plate reader. Enzyme activity units are defined as the amount of nmol *p*NP released by 1 μL of culture supernatant in 1 h. β-glucosidase activity was detected using *p*-nitrophenyl-β-D-glucopyranoside, and β-xylosidase activity was detected using *p*-nitrophenyl-β-D-xylopyranoside (both from Macklin, Shanghai, China). The *p*-nitrophenol linked substrates were prepared as 50 mM stock solutions in water. Two technical replicates and three biological replicates were set up in the experiment. In a time-course analysis of the enzyme activities, aliquots from 0 h, 12 h, 24 h, and 36 h, from D-fructose, no carbon, and cellulose cultures were analyzed.

Cellulase activity assays were used to measure the breakdown of cellulose by enzymes from the culture supernatants. The assays were performed in 2 mL tubes using a total volume of 750 μL containing 50 mM sodium acetate pH 4.5, 10 mg (1.3% *w*/*v*) cellulose (microcrystalline from cotton linters) with 100 μL culture supernatant from separate cellulose shake-flask cultures from 18 h or 36 h, and 0.02% *w/v* NaN_3_ to prevent microbial growth. The reactions were shaken horizontally at 37 °C with 200 RPM for up to 72 h. At 0 h, 6 h, 24 h, and 72 h, aliquots of 70 µL were taken from the reactions and were heated at 99 °C for 5 min to inactivate the enzymes. The glucose concentration was measured using a D-Glucose Assay Kit (GOPOD Format) (Megazyme, Bray, Ireland) by measuring the absorbance at 490 nm using a spectrophotometric plate reader.

For PAGE gel analysis, the culture supernatants were concentrated using 10 kDa MWCO diafiltration columns (Millipore, Cork, Ireland) to 10–20 times the original concentration. The protein samples were mixed with 5X loading buffer containing DTT (Cat. no P0015L, Beyotime, Shanghai, China) and heated at 95 °C for 5 min. Sodium dodecyl sulfate-polyacrylamide gel electrophoresis (SDS-PAGE) was performed using a 10% (*w*/*v*) polyacrylamide gel prepared according to Ultrafast SDS-PAGE Gel Preparation Kit (Zomanbio, Beijing, China). A pre-stained protein ladder (Solarbio, Beijing, China) was also run on the PAGE gels. The PAGE gel was run using a Mini-PROTEAN Tetra Cell System (Bio-Rad, Hercules, CA, USA) with running buffer containing 0.1% *w*/*v* SDS, 0.303% *w*/*v* Tris base, and 1.44% *w*/*v* glycine. The PAGE gels were silver-stained according to the Fast Silver Stain Kit (Beyotime).

### 2.6. RNA Extraction, Sequencing, and Analysis of the T. harzianum Δxyr1 Mutant

Total RNA was extracted using the Magnetic Tissue/Cell/Blood Total RNA Kit (Tiangen Biotech, Beijing, China) according to the manufacturer’s instructions. RNA purity and quantity were evaluated using a NanoDrop spectrophotometer (Thermo Scientific, Waltham, MA, USA), and RNA integrity was evaluated using the Agilent 2100 Bioanalyzer (Agilent Technologies, Santa Clara, CA, USA).

For the library preparation for transcriptome sequencing, a total amount of 1.5 µg RNA per sample was used as input material for the RNA sample preparations. Sequencing libraries were generated using NEBNext^®^ Ultra™ RNA Library Prep Kit for Illumina^®^ (NEB, Ipswich, MA, USA) following the manufacturer’s recommendations, and index codes were added to attribute sequences to each sample. Briefly, mRNA was purified from total RNA using poly-T oligo-attached magnetic beads. Fragmentation was carried out using divalent cations under elevated temperature in NEBNext First Strand Synthesis Reaction Buffer (5X). To select cDNA fragments of preferentially 200–250 bp in length, the library fragments were purified with the AMPure XP system (Beckman Coulter, Brea, California, USA). Then, 3 μL USER Enzyme (NEB, USA) was used with size-selected, adaptor-ligated cDNA at 37 °C for 15 min, followed by 5 min at 95 °C, before PCR was performed with Phusion High-Fidelity DNA polymerase, Universal PCR primers, and Index (X) Primer. The purified PCR products (AMPure XP system) and library quality was assessed on the Agilent Bioanalyzer 2100 system. The library preparations were sequenced on an Illumina NovaSeq 6000 platform by Beijing Allwegene Technology (Beijing, China), and paired-end 150 bp reads were generated. About ~40 M raw reads for each sample were generated. The RNA-Seq reads from this project were submitted to the GEO database (GEO accession GSE252008).

Raw data (raw reads) of fastq format were first processed through in-house Perl scripts. In this step, clean data (clean reads) were obtained by removing reads containing adapter sequences, reads containing poly-N, and low-quality reads from raw data. At the same time, Q20, Q30, GC-content, and the sequence duplication level of the clean data were calculated. All downstream analyses were based on clean data with high quality. These clean reads were then mapped to the reference genome sequence using STAR aligner (v2.5.2b) [29]. Only reads with a perfect match or one mismatch were further analyzed and annotated based on the reference genome for the *T. harzianum* CBS 226.95 strain from NCBI at https://www.ncbi.nlm.nih.gov/datasets/genome/GCF_003025095.1/ (accessed on 13 November 2023) [30]. The percentage of uniquely mapped reads was at least 95% in all samples. HTSeq (v 0.5.4) [31] was used to count the read numbers mapped to each gene. Gene expression levels were estimated by fragments per kilobase of transcript per million fragments mapped (FPKM). Differential expression analysis was performed using the DESeq2 (1.14.1) [32]. The general criteria for a gene to be considered differentially expressed were >2-fold change in expression and P_adj_ < 0.05 from DESeq2 analysis, and an FPKM > 1 in one condition. The PCA was performed using FactoMineR [33] in the R statistical environment. Hierarchical clustering was performed using the log_2_ FPKM values in the R statistical environment, using the gplots package with the Euclidian distance and complete linkage options selected. The Venn diagrams were generated using the EVenn tool at http://www.ehbio.com/test/venn/#/ (accessed on 13 November 2023) [34].

For gene annotations of the *T. harzianum* Triha1 genes, the annotations from JGI Mycosm were used [35]. For the assignment of the plant biomass-degrading subset of CAZymes and activities, the annotations from [30] were used. For the assignments of *T. harzianum* orthologs of transcription factors related to CAZyme regulation, and the assignment of orthologs of carbon metabolism pathway genes, the annotations of *T. reesei* QM6a Trire2 genes from [20] were assigned to the *T. harzianum* reciprocal BLAST best hit. The reciprocal BLAST best hit analysis of Trire2 and Triha1 proteins was performed in Galaxy using BLAST Reciprocal Best Hits (Galaxy Version 0.3.0) [36,37] using the UseGalaxy.eu server [38]. Appendix A lists the annotations used in the analysis.

## 3. Results

### 3.1. T. harzianum xyr1 Deletion Mutants Generated Using Hygromycin Resistance Cassette

The *T. harzianum* CBS 226.95 is one of the most widely studied *T. harzianum* strains; it is considered the reference strain for the species, and was used here for the deletion of *xyr1*. A linear fragment containing a hygromycin resistance cassette flanked by sequences upstream and downstream of *T. harzianum xyr1* (*Triha1_1126*) was used for the deletion of *xyr1*. Two independent deletion mutants were purified through single-spore isolation after the initial screen of the transformants (Δ*xyr1-2* and Δ*xyr1-63*). Two approaches demonstrated that the mutants had the *xyr1* gene deleted. Firstly, no PCR product for the *xyr1* coding sequence was obtained from the deletion mutants from the attempted amplification using *xyr1* coding sequence primers, as Figure 1B demonstrates. The primers for *xyr1* amplified well from the wild type gDNA, and also, the gDNA from the mutant was of good quality because there was a PCR product amplified using primers for *tef1* (translation elongation factor 1-alpha) and *hph* (hygromycin phosphotransferase) (Figure 1B). Secondly, another set of primers was used that bound to the DNA outside of the sequences flanking the resistance cassette to verify the recombination of the deletion cassette at the *xyr1* locus (see Figure 1A for the location of these primers). The difference in the size of the products amplified from the wild type and Δ*xyr1* loci was too small to easily distinguish on an agarose gel, and instead, restriction digests were used to distinguish the two PCR products. The *KpnI* restriction enzyme digested twice in the product amplified from the Δ*xyr1* locus and did not digest in the product amplified from the wild type locus, thus confirming the recombination of the hygromycin resistance cassette at the *xyr1* locus in both Δ*xyr1* strains (Figure 1C).

### 3.2. The T. harzianum Δxyr1 Mutant Showed Reduced Growth on a Subset of Carbon Sources

For plate growth profiling, two independent *T. harzianum* Δ*xyr1* mutant strains (Δ*xyr1-2* and Δ*xyr1-63*) were grown on eight different carbon sources, including monosaccharides, a disaccharide, and polysaccharides, and compared with the parental wild-type strain (Figure 2). The growth was compared based on the colony diameter and the density of the colony appearance after 48 h, and based on the density of the colony appearance and apparent sporulation levels at 96 h. As expected for a deletion of *xyr1*, there were no clear, consistent differences between the two Δ*xyr1* mutants and wild type on D-glucose, D-fructose, and without an added carbon source, in terms of colony diameter, colony density, or levels of visible sporulation (Figure 2 and Appendix A).

Cellulose and xylan are two key components of plant biomass, and growth on these polymers, and sugars that compose these polymers (i.e., cellobiose from cellulose, and D-xylose and L-arabinose from xylan) can indicate the role of *T. harzianum* XYR1 in the degradation and catabolism of plant biomass-related polysaccharides. At 48 h and 96 h on cellulose, the Δ*xyr1* mutant mycelia appeared less dense than the wild type, and there was less sporulation in the Δ*xyr1* mutants at 96 h, although there was also no significant decrease in the radial growth at 48 h (Appendix A). On cellobiose, there was also no clearly visible decrease in the growth of the Δ*xyr1* mutants compared to the wild type. At 48 h on D-xylose, there was a significant reduction (*p* < 0.05) of 15–25% in the colony diameter of both Δ*xyr1* mutants, along with a less dense mutant colony appearance (Figure 2). At the 96 h time-point, as well as a less dense colony appearance, there was much less sporulation visible on the Δ*xyr1* mutant cultures on D-xylose (Appendix A). The appearance of the Δ*xyr1* mutant colonies on D-xylose was similar to their appearance of the cultures without an added carbon source, suggesting that the Δ*xyr1* mutants cannot catabolize D-xylose for any level of growth. On L-arabinose, there appeared to be a small reduction in sporulation levels at the 96 h time-point in the Δ*xyr1* mutants compared to the wild type, but there were no clear, consistent reductions in colony diameter or density in the Δ*xyr1* mutants; however, it needs to be stated that the wild type grew relatively poorly on L-arabinose compared to the other monosaccharides tested. On xylan from corn cob, there was also a less dense colony appearance at 48 h but not to the same extent as on D-xylose, and there were no significant reductions in colony diameter compared to the wild type. At the 96 h time-point, the mutant colonies grown on xylan were also less dense than the wild type, and less sporulation was visible.

The reduction in growth on the polysaccharides suggested a reduction in enzymatic activity in the Δ*xyr1* mutant, and the enzymatic activities related to the degradation of cellulose and xylan polysaccharides were measured. As both *T. harzianum* Δ*xyr1* mutants showed the same growth trends on the different carbon sources, only one of the mutants (Δ*xyr1-2*) was used for subsequent analyses.

### 3.3. Enzyme Activity and Protein Secretion Show Major Decreases in the T. harzianum Δxyr1 Mutant

We analyzed the enzyme activity of the *T. harzianum* Δ*xyr1-2* mutant, as compared to the *T. harzianum* wild-type strain by measuring β-glucosidase, β-xylosidase, and cellulase activities from culture supernatants at various time-points from no carbon, D-fructose, and cellulose shake-flask cultures (Figure 3).

The presence of cellulose clearly induced β-glucosidase and β-xylosidase activities, with activities detected at 12 h after the transfer of mycelia from the *T. harzianum* D-fructose pre-cultures to shake-flasks containing cellulose (Figure 3), whereas there was little or no β-glucosidase and β-xylosidase activity detected at the same time-points after transfer to D-fructose cultures or cultures without a carbon source. At 12 h in the cellulose wild-type cultures, the β-glucosidase activity and β-xylosidase activity were detected at ~0.2 nmol/h/µL, and were maintained at a similar level at 24 h and 36 h. In contrast, the *T. harzianum* Δ*xyr1-2* β-glucosidase activity was ~90% lower than the wild type, and β-xylosidase was not detected in the Δ*xyr1-2* mutant. Clearly, the transcriptional activator XYR1 is crucial for β-xylosidase and β-glucosidase production in *T. harzianum* cellulose cultures. From the activity of β-glucosidase and β-xylosidase, 18 h was selected as the time-point for the RNA-Seq. The 18 h time-point was analyzed for cellulase activity and PAGE gel analysis alongside the 36 h time-point samples.

Cellulase activity was measured from *T. harzianum* wild type and Δ*xyr1-2* 18 h and 36 h cellulose-induced cultures. The amount of glucose released from the cellulose was measured at three time-points in the enzyme assay (6 h, 24 h, and 72 h). The same volumetric amount of culture supernatants were used from the 18 h and 36 h cultures, and there was higher cellulase activity from the 36 h compared to the 18 h cultures. Approximately 10% cellulose was converted to glucose using the enzymes from the 36 h wild-type culture. In stark contrast, no cellulase activity was detected from the Δ*xyr1-2* mutant cultures compared to the wild type from neither the 18 h nor the 36 h cultures. The detection limit of the glucose measurement assay was 33.3 µg/mL in the enzyme reaction or equivalent to the conversion of 0.25% of the cellulose to glucose. Even though more cellulase activity was detected from the 36 h wild-type cultures compared to the 18 h wild-type cultures, no cellulase activity was detected from either of the Δ*xyr1-2* mutant culture time-points. The lack of glucose detection did not change over hydrolysis time, highlighting that cellulase activity from the Δ*xyr1-2* mutant culture was very low. The large decrease in activity between the wild type and the Δ*xyr1-2* mutant from 18 h and 36 h suggests that XYR1 is the major regulator of cellulase activity when cellulose is the inducer and carbon source.

The secreted proteins from the 18 h and 36 h *T. harzianum* wild-type and Δ*xyr1-2* mutant cultures were analyzed using PAGE gel (Figure 3). In the protein samples from wild-type cultures, the banding patterns clearly showed an induction of protein production due to the presence of cellulose, as several bands were present in the supernatant from the wild-type cellulose cultures that were absent from the supernatants from the D-fructose and no carbon control cultures. In stark contrast to the wild-type cellulose cultures, almost no protein bands were detected from the Δ*xyr1-2* mutant control cultures even though sensitive silver staining was used. The absence from the Δ*xyr1-2* mutant supernatants of many of the protein bands visible in the wild-type cellulose culture supernatants is consistent with the low level or lack of β-glucosidase, β-xylosidase, and cellulase enzymatic activities detected from the Δ*xyr1-2* mutant.

### 3.4. Overview RNA-Seq Analysis of the T. harzianum Δxyr1 Mutant

To investigate cellulose-induced genes that were regulated by *T. harzianum* XYR1, transcriptomics was used. At 18 h after transfer to shake-flask cultures, the transcriptome on the wild type and the Δ*xyr1-2* mutant growing on cellulose was compared, and also compared to D-fructose control cultures, which identified which genes were induced by cellulose. The transcriptome analysis of a further control without a carbon source was compared to show whether the stress condition of the absence of a carbon source may have also regulated genes. Notably, when mycelial samples from the 18 h shake-flask cultures were collected, fewer mycelia were visible in the cellulose cultures of the Δ*xyr1-2* mutant compared to the wild-type cellulose cultures.

In the PCA analysis of the expression of all genes, there was a clear separation of the expression of the wild-type cultures on cellulose compared to the Δ*xyr1-2* mutant on cellulose (Figure 4A). Interestingly, the transcriptomes of the Δ*xyr1-2* mutant on cellulose clustered with all of the no carbon cultures from either the wild type or the Δ*xyr1-2* mutant cultured without a carbon source. This clustering pattern suggested the major effects of the deletion of *xyr1* on the physiology and ability of the *T. harzianum* Δ*xyr1-2* mutant to grow on cellulose, to the extent that the transcriptomes are similar to that of *T. harzianum* cultured in the absence of a carbon source. The PCA analysis showed that the major effect of the deletion of *xyr1* was when cultured with cellulose, whereas the wild-type and Δ*xyr1-2* culture replicates growing on D-fructose clustered among each other, and the wild-type and Δ*xyr1-2* culture replicates on no carbon also clustered among each other (Figure 4A). There was also a separation in the clusters of wild type cultured with cellulose and wild type cultured with D-fructose, albeit the distance for one of the replicates was less than between other groups but showed a clear effect of cellulose on the transcriptome of *T. harzianum*.

The patterns in the PCA were also reflected in the number of DE genes (Figure 4B). In the comparison of the wild type and the Δ*xyr1-2* mutant cultured on cellulose, there were 5168 genes differentially expressed, with similar numbers of genes with a lower and with a higher expression compared to the mutant. There were 1660 genes differentially expressed in the comparison of the wild type on cellulose and the wild type on D-fructose, with 719 genes higher expressed on cellulose compared to the D-fructose cultures. Of these 719 genes with a higher expression on cellulose in the wild type, ~75% had a lower expression in the Δ*xyr1-2* mutant, and these genes were considered to be directly or indirectly regulated by XYR1 (Figure 4C). Very few genes were differentially expressed in the comparisons of the wild-type culture on no carbon, the Δ*xyr1-2* culture on cellulose, and the Δ*xyr1-2* culture on no carbon, which was consistent with the clustering pattern of the PCA. Also, there were very few genes differentially expressed in the comparison of the wild-type and Δ*xyr1-2* cultures on D-fructose, which supports the lack of background or other mutations besides Δ*xyr1* in the mutant, and is in line with the lack of differences in the growth profiling in plate cultures containing D-fructose.

### 3.5. RNA-Seq Analysis of the T. harzianum Δxyr1 Mutant Shows the Regulation of Xylanases as well as Cellulases

The *T. harzianum* genome encodes for 128 plant biomass-degrading (PBD) CAZymes, and the summing of the total expression of the PBD CAZymes showed that there was an ~8-fold induction of these PBD CAZymes in the wild type on cellulose compared to the D-fructose cultures, to a level of expression that was 2.6% of the total expression. In the Δ*xyr1-2* mutant on cellulose compared to the wild type on cellulose, there was a huge 81% reduction in the total expression of PBD CAZymes (Figure 5).

There were 50 PBD CAZymes that had a lower expression in the Δ*xyr1-2* mutant on cellulose compared to the wild type on cellulose. Of these PBD CAZymes, 43 were induced by cellulose (i.e., significantly higher expressed in the wild-type cultures on cellulose compared to D-fructose). There were ~30 PBD CAZymes whose expression in the Δ*xyr1-2* mutant on cellulose was reduced to a similar level to the wild type or the Δ*xyr1-2* mutant on D-fructose, suggesting that the induction on cellulose of these PBD CAZymes was completely regulated by XYR1. Twenty-one PBD CAZymes had a higher expression in the Δ*xyr1-2* mutant on cellulose compared to the wild type on cellulose, and this may partly be explained by a carbon starvation-type response as 18 of these had a higher expression in the Δ*xyr1-2* mutant on no carbon compared to the wild type growing on cellulose. The hierarchical clustering in Figure 5A gives an overview of the expression pattern changes in PBD CAZymes.

There are 36 PBD CAZy genes in *T. harzianum* that are predicted to have an activity involved in the degradation of cellulose, the polymer present in the shake-flask cultures used for RNA-Seq analysis. The total expression of PBD CAZy cellulases in the wild-type cultures increased ~20-fold from ~600 FPKM on D-fructose to ~16,700 FPKM on cellulose, and the total expression was only ~700 FPKM in the Δ*xyr1-2* mutant on cellulose (Figure 5). Of the 14 endo-β-1,4-glucanases, 6 were induced by the presence of cellulose, and all of these had a lower expression in the Δ*xyr1-2* mutant on cellulose compared to the expression in the wild type on cellulose. There are two cellobiohydrolases in *T. harzianum,* and both appear to be completely regulated by XYR1 because their expression was <5 FPKM in the Δ*xyr1-2* mutant on cellulose compared to ~5000 FPKM in the wild type on cellulose. There are 16 *T. harzianum* genes predicted as β-glucosidases, and 8 were induced by the presence of cellulose, and 7 had a lower expression in the Δ*xyr1-2* mutant on cellulose. Another two of the *T. harzianum* β-glucosidases (GH3 family members *Triha1_504610* and *Triha1_96503*) did not have a lower expression in the Δ*xyr1-2* mutant and appeared to be constitutively expressed as they were also expressed at similar levels on the D-fructose and no carbon cultures. There are three AA9 family LPMOs in *T. harzianum*, and two of these appeared to be completely regulated by XYR1 with an expression of ~1000 FPKM in the wild-type cellulose cultures, and almost zero in the Δ*xyr1-2* mutant on cellulose, while the third AA9 family LPMO was not expressed in any of the wild-type or mutant cultures.

*T. harzianum* contains 35 genes predicted to be active on xylan, including main-chain acting endo-xylanases and β-xylosidases, and side-chain acting α-arabinofuranosidases and α-glucuronidases, and esterases. The total expression of PBD CAZy predicted to be active on xylan in the wild-type cultures increased 12-fold from ~500 FPKM on D-fructose to ~6000 FPKM on cellulose, and the total expression was only ~500 FPKM in the Δ*xyr1-2* mutant on cellulose. Of the nine *T. harzianum* endo-xylanases (from GH10, GH11, and GH30_7 families), eight were expressed >1 FPKM, and all eight of these were induced in wild-type cellulose cultures, and appeared to be almost completely regulated by XYR1 as the expression level in the Δ*xyr1-2* mutant on cellulose was <5 FPKM, which was similar to the expression levels for the genes in the D-fructose and no carbon cultures. Of the six predicted β-xylosidases in *T. harzianum*, three were induced in cellulose cultures, and all three had a lower expression in the Δ*xyr1-2* mutant on cellulose, indicating that these were also regulated by XYR1. Also, nine of the predicted xylan side-chain acting enzymes were induced in cellulose cultures and all nine had a lower expression in the Δ*xyr1-2* mutant. *T. harzianum* also contains PBD CAZymes that act on other polysaccharides such as pectin, xyloglucan, and mannan, and subsets of these enzymes were also induced by cellulose in wild-type cultures, and had a lower expression in the Δ*xyr1-2* mutant, indicating that they are also regulated by XYR1 (Figure 5 and Appendix A).

### 3.6. Expression Changes in the Δxyr1 Mutant beyond Plant Biomass-Degrading CAZymes

A list of putative *T. harzianum* carbon catabolic enzymes was made from the reciprocal best BLAST hits of carbon catabolic enzymes annotated in *T. reesei* [20] (Appendix A). Of the putative carbon catabolic enzymes with a higher expression in wild-type cellulose cultures, these included most of the genes that could function on a pentose catabolic pathway (*T. harzianum* orthologs of *lad1*, *xdh1*, *xki1*, and *xyl1*), and a D-galactose-oxido-reductive pathway (*T. harzianum* orthologs of *lad1*, *xdh1*, *xyl1*, and *lxr4*). In the Δ*xyr1-2* mutant on cellulose, the *xyl1* and *xki1* orthologs had a significantly lower expression compared to the wild type on cellulose, while *xdh1* was not significantly different, and *lad1* had a higher expression in the Δ*xyr1-2* mutant on cellulose. The higher expression of *lad1* in the Δ*xyr1-2* mutant on cellulose may be related to carbon starvation because the *lad1* expression was also higher in the no carbon wild type compared to D-fructose wild type cultures.

The expression of other transcription factors when *xyr1* is deleted can indicate whether the changes in expression are likely directly related to *xyr1* deletion or indirectly due to changes in the expression of other transcription factors, and probably the genes regulated by those other transcription factors. *T. harzianum xyr1* was induced ~5-fold in the cellulose wild-type cultures compared to D-fructose control, and as expected, the expression was not detected from the Δ*xyr1-2* mutant samples. A list of putative *T. harzianum* carbon utilization-related transcription factors was made from reciprocal best BLAST hits from the transcription factor orthologs in *T. reesei* that consisted of *xpp1*, *cre1*, *mcmA*, *ace1*, *bglR*, *clbr2*, *ara1*, *ace3*, *amyR*, *gaaR*, *malR*, *rhaR*, *gaaX*, *ace2*, *rce1*, *clbr3*, and *clr2* (Appendix A). As well as *xyr1*, the *T. harzianum* orthologs of the two cellulase activators *ace3* and *clr2* were higher expressed in the wild type cellulose compared to D-fructose cultures, and both *ace3* and *clr2* had a lower expression in the Δ*xyr1-2* mutant on cellulose. Although *xpp1*, *cre1*, and *amyR* transcription factors did not have a higher expression on wild type cellulose compared to D-fructose cultures, all three of these transcription factors had a lower expression in the Δ*xyr1-2* mutant on cellulose compared to wild type on cellulose. The transcription factors *rce1* and *clbr3* had a higher expression in the Δ*xyr1-2* mutant on cellulose compared to the wild type on cellulose. The other ten transcription factor orthologs analyzed (*mcmA*, *ace1*, *bglR*, *clbr2*, *ara1*, *gaaR*, *malR*, *rhaR*, *gaaX*, and *ace2*) were all expressed > 1 FPKM, but none were higher expressed on wild type cellulose compared to D-fructose cultures nor differentially expressed between the wild-type and Δ*xyr1-2* mutant cellulose cultures (Appendix A).

Sugar transport is another key component of plant biomass utilization, and there are 90 *T. harzianum* genes annotated with the Pfam domain for sugar (and other) transporter (PF00083) (Appendix A). Of these predicted transporters, 11 were higher expressed on the cellulose compared to D-fructose wild-type cultures, and 9 of these were lower expressed on Δ*xyr1-2* compared to wild-type cellulose cultures. There were five other sugar transporters with a lower expression in the Δ*xyr1-2* mutant compared to wild type on cellulose that were not induced by the presence of cellulose in the wild-type cultures. There were 28 sugar transporters that had a higher expression in the Δ*xyr1-2* mutant compared to wild type on cellulose, and the higher expression of about half of these may be related to carbon starvation conditions as the expression of 13/28 was higher also in the wild-type no carbon condition compared to the wild-type D-fructose condition (Appendix A).

## 4. Discussion

Here, we have demonstrated how the *T. harzianum* transcriptional activator XYR1 affects growth and enzymatic activities, regulates a wide range of plant biomass-degrading CAZymes, and appears to be the major regulator for the degradation of cellulose, with profound adverse stress effects in Δ*xyr1* deletion mutants cultured with cellulose.

The reduced growth of the *T. harzianum* Δ*xyr1* mutants on D-xylose is similar to the reduced growth of the *T. reesei* QM9414 Δ*xyr1* mutant on D-xylose [9,20]. Notably, the growth appears a lot less for the *T. reesei* QM9414 Δ*xyr1* mutant compared to the *T. harzianum* Δ*xyr1* mutants, and this is likely due to the greater ability of *T. harzianum* to grow on agarose as a sole carbon source. The *T. harzianum* Δ*xyr1* mutant is the second species within the *Trichoderma* genus, where reduced growth is demonstrated on D-xylose. Although there are Δ*xyr1* mutants in *T. atroviride* [24] and *T.* cf. *guizhouense* [23], to our knowledge, there are no tests of growth on D-xylose in these studies. The reduced growth of the two *Trichoderma* species Δ*xyr1* mutants on D-xylose is similar to the reduced growth on D-xylose in *Fusarium graminearum* Δ*xyr1* and *Magnaporthe oryzae* Δ*xlr1*, but in stark contrast to two Aspergilli species, where in *A. nidulans* and *A. niger* Δ*xln*R mutants, there is no reduction in growth on D-xylose [39]. A role for *T. harzianum* XYR1 in regulating D-xylose catabolism is supported by the lower expression of two genes (*xyl1* and *xki1*) putatively from the pentose catabolic pathway in the Δ*xyr1-2* cultures on cellulose compared to wild-type cultures on cellulose (Appendix A). There was clearly reduced growth on corn cob xylan in the Δ*xyr1* mutants. This could be explained by the reduced ability of the Δ*xyr1* mutants to grow on xylose, which is one of the main components of xylan, and also the lower expression of xylan-degrading enzymes such as β-xylosidases and endo-xylanases which the RNA-Seq data from the cellulose cultures showed were regulated by XYR1. Xylan extracted from corn cobs was shown to have a sugar composition that, as well as xylose, included arabinose and glucose [40], and the presence of these other sugars may account for some of the growth of the Δ*xyr1* mutants on corn cob xylan.

There was no clear reduction in the growth of the Δ*xyr1* mutants on cellobiose (Figure 2 and Appendix A), and this was surprising because the RNA-Seq data from the cellulose cultures showed that several β-glucosidases, that hydrolyze cellobiose to glucose, were regulated by XYR1, but there were also at least two other β-glucosidases that were expressed and not regulated by XYR1 (Appendix A), and β-glucosidase activity was detected from the Δ*xyr1* culture supernatants (Figure 3). Perhaps, the β-glucosidase activity from these other non-XYR1-regulated β-glucosidases is sufficient to hydrolyze the cellobiose for growth in the plate cultures. Although cellobiose was not used as a carbon source in the shake-flask cultures, it is interesting to speculate on what might be the enzymatic and secretory response of the wild-type and Δ*xyr1* strains to cellobiose. There are two studies suggesting that cellobiose is a relatively poor inducer of cellulolytic activities in *T. harzianum*. In the *T. harzianum* FJ1 strain cultured with cellobiose compared to cellulose, the levels of cellulase and xylanase, but not β-glucosidase, activities were much lower [41]. In the *T. harzianum* KUC1716 strain, the authors stated that the endoglucanase activity was low when cellobiose was used as the inducer compared to cellulose [42]. It is also possible that when cellobiose is used as the inducer, there could be some carbon catabolite repression effects due to an excess of glucose released from cellobiose. In any event, as the cellulolytic activity is likely relatively lower when cellobiose is the inducer, the effect of the deletion of *xyr1* is also likely to be less compared to when other carbon sources such as cellulose are used as the inducer of cellulolytic activity. On the cellulose plate cultures, the reduced growth of the Δ*xyr1* mutants is likely explained by the hugely reduced expression of endo-cellulases and cellobiohydrolases in the Δ*xyr1-2* mutant in cellulose shake-flask cultures (Figure 5), as well as the inability of the culture supernatants from the Δ*xyr1-2* mutant cellulose cultures to release any detectable glucose from the cellulose in the cellulase enzymatic assays (Figure 3B).

In an analysis of XYR1/XlnR/XLR-1 mutants in different species, one of the key differences between species is whether cellulases, xylanases, or both are regulated by XYR1/XlnR/XLR-1. Recently, in *Myceliophthora thermophila*, the deletion of *xlr1* was shown to reduce mainly xylanase gene expression when xylose or xylan was the inducer, while cellulase gene expression was not altered [43], similar to *N. crassa* XLR-1 [10]. *T. harzianum* XYR1 appears to regulate both cellulase and xylanase genes based on the large reductions in expression of most cellulase and xylanase genes in the Δ*xyr1-2* mutant in the cellulose shake-flask cultures (Figure 5), and the reduced growth on both xylan and cellulose plate cultures (Figure 2 and Appendix A). In *T.* cf. *guizhouense* NJAU4742, when Avicel cellulose was used as the inducer, both cellulases and xylanases were induced, and both sets of genes had a much lower expression in the *T.* cf. *guizhouense* Δ*xyr1* mutant [23]. The expression pattern in the *T. harzianum* Δ*xyr1* mutant on cellulose appears consistent with the *T.* cf. *guizhouense* Δxyr1 mutant results. The analysis of C-catabolic enzymes and growth on various carbon sources was not analyzed with the *T.* cf. *guizhouense* Δ*xyr1* mutant, and therefore is not possible to compare with the data from our *T. harzianum* Δ*xyr1* mutant. The induction of xylan-degrading enzymes and D-xylose catabolic enzymes by cellulose may be somewhat surprising, but breakdown products of cellulose, such as cellobiose, may also lead to the activation of xylan-degrading enzymes. Alternatively, the microcrystalline cellulose derived from cotton linters used in the cultures may contain small amounts of xylan or xylose, which could contribute to the induction of the xylan-degrading enzymes.

Another key factor in comparing *xyr1* mutants in different species and carbon sources is the extent to which XYR1 controls the regulation of the degradation of a particular substrate. One of the remarkable results was how the transcriptomes of the *T. harzianum* Δ*xyr1* mutant on cellulose clustered with transcriptomes of the cultures without an added carbon source (Figure 4). This pattern suggests that *T. harzianum* XYR1 almost completely controls the degradation of cellulose in shake-flask cultures. There were similar suggestions of stress responses in *T. reesei* RUT-C30 Δ*xyr1* analysis whereby starvation stress-related genes had a higher expression in the *T. reesei* RUT-C30 Δ*xyr1* mutant on lignocellulose-related substrates [22].

Previously, *xyr1* was overexpressed in the *T. harzianum* strain P49P11 [25], and it is useful to compare the trends in the expression analysis of selected genes in the *xyr1* overexpression mutant with the expression of those genes in the RNA-Seq dataset with the Δ*xyr1-2* mutant. The gene IDs were not listed in that study but using the primer sequences from that study [25], the Triha1 gene IDs were found using the PrimerBLAST software at NCBI whereby the software works by searching a database of Triha1 sequences for matches with the primer sequences. The following six genes all had a higher expression in the *xyr1* overexpression strain in the sugarcane bagasse cultures: *cbh1* (Triha1_7497), *CE5* (Triha1_118643), *cre1* (Triha1_502975), *EGIII* (Triha1_91916), *GH10* (Triha1_91773), and *GH11* (Triha1_115099). The trend from the *xyr1* overexpression strain was the inverse of the trend in the Δ*xyr1-2* deletion mutant cultures on cellulose, whereby all seven of these genes had a significantly lower expression (2FC, P_adj_<0.05, FPKM>1) compared to the wild-type cultures on cellulose. It was not possible to correlate the fold changes in expression because for *cbh1* (Triha1_7497), *CE5* (Triha1_118643), *EGIII* (Triha1_91916), *GH10* (Triha1_91773), and *GH11* (Triha1_115099), the expression was close to zero FPKM in the Δ*xyr1-2* mutant culture on cellulose (Appendix A). Also, the higher expression of c*re1* in the *xyr1* overexpression mutant [25] possibly led to some repression effects, which may confound attempts to correlate the expression levels in the datasets from the overexpression and deletion mutants.

The *T. harzianum* Δ*xyr1* deletion mutant is a useful resource for future investigations on the regulation of transcriptional responses to crude plant biomass substrates, and investigating which sugars are signaling via XYR1. Also, investigating the role of XYR1 in induced resistance in plants has potential for strain improvement in *T. harzianum* for biocontrol applications.

## Figures and Tables

**Figure 1 biomolecules-14-00148-f001:**
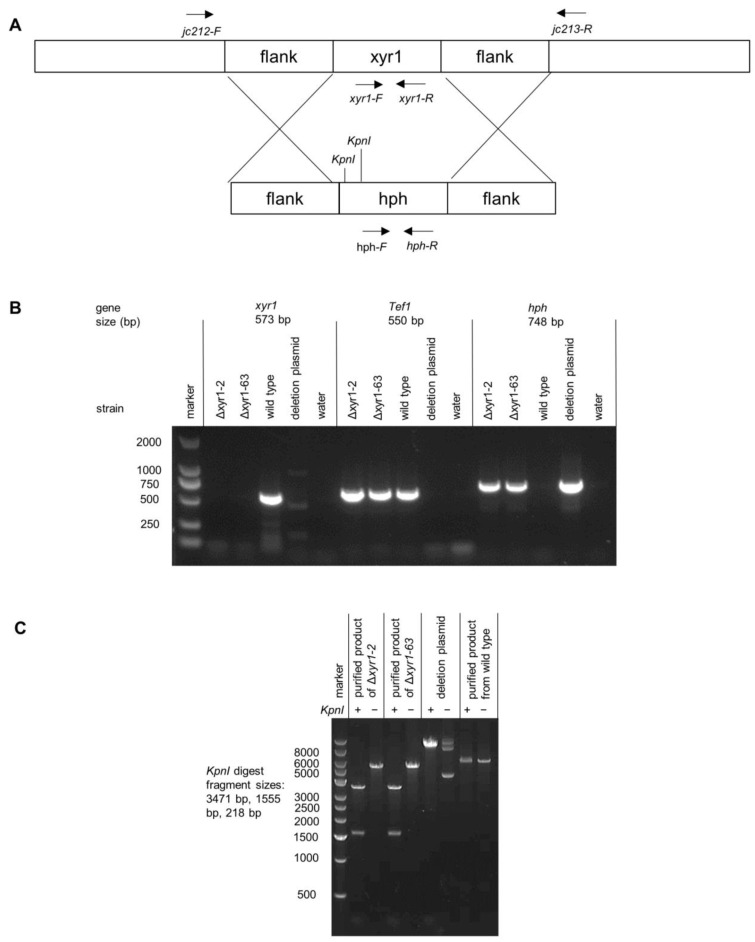
PCR and restriction digest verification of *T. harzianum* Δ*xyr1* mutants. (**A**) Schematic diagram of *xyr1* locus and deletion cassette highlighting the location with black arrows of primer binding and restriction digest sites, (**B**) agarose gel images demonstrating the absence of the *xyr1* sequence in the mutants, and (**C**) gel image from the restriction digest of the PCR product that spans across the *xyr1* locus to verify integration at the *xyr1* locus. It should be noted that the PCR products from wild type and Δ*xyr1* loci were too similar in size to distinguish, and instead, restriction digests of the products were used.

**Figure 2 biomolecules-14-00148-f002:**
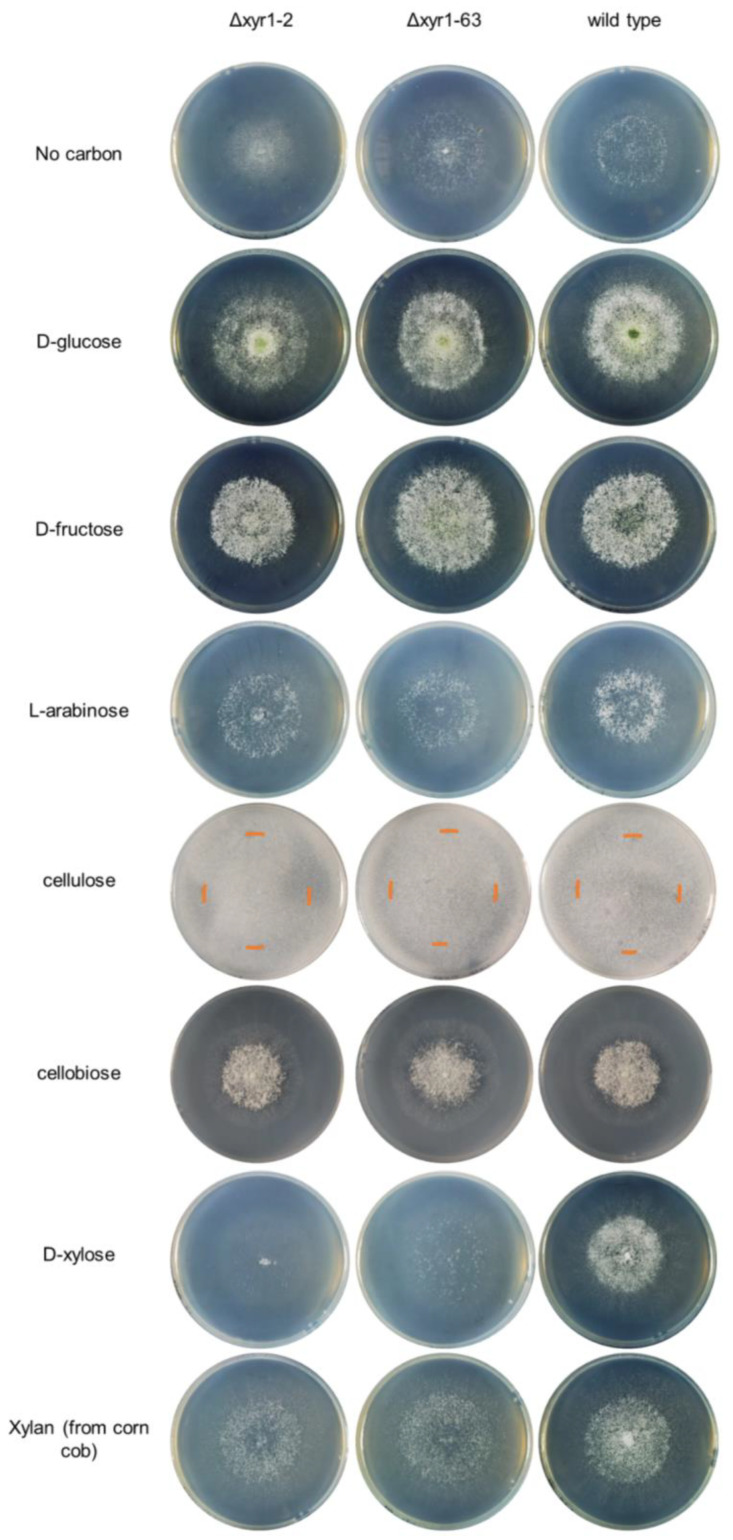
Growth profiling analysis of *T. harzianum* Δ*xyr1* deletion mutants. The edge of the colony on the cellulose cultures is marked with an orange-colored line on the plate images. The growth profiling experiment was repeated twice with similar results from both repeats. Appendix A contains the colony diameter measurements from the two repeat experiments.

**Figure 3 biomolecules-14-00148-f003:**
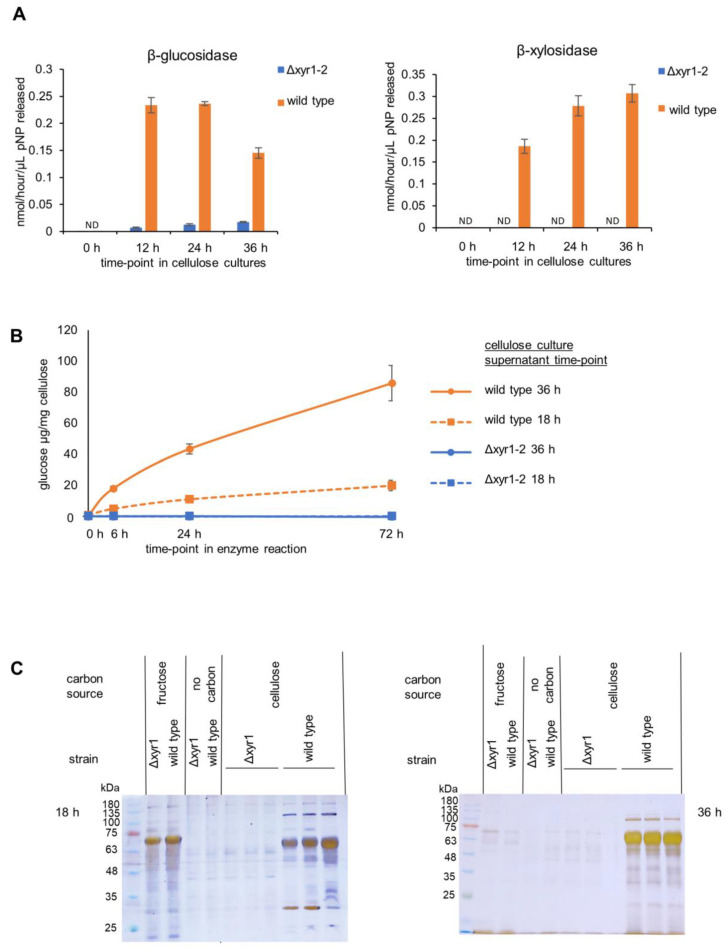
Enzyme activity and PAGE gel analysis of the *T. harzianum* Δ*xyr1* mutant. (**A**) β-glucosidase and β-xylosidase enzyme activities from the time-course of wild-type and Δ*xyr1* mutant cellulose cultures, (**B**), cellulase activity from wild-type (orange-colored lines) and Δ*xyr1* mutant (blue-colored lines) 18 h (dashed lines) and 36 h (solid lines) cellulose-induced cultures, and (**C**) silver-stained PAGE gel analysis of supernatants from wild-type and Δ*xyr1* mutant 18 h and 36 h cellulose cultures. The cellulase activity assay was repeated twice with the same trends in both repeats. ND = not detected.

**Figure 4 biomolecules-14-00148-f004:**
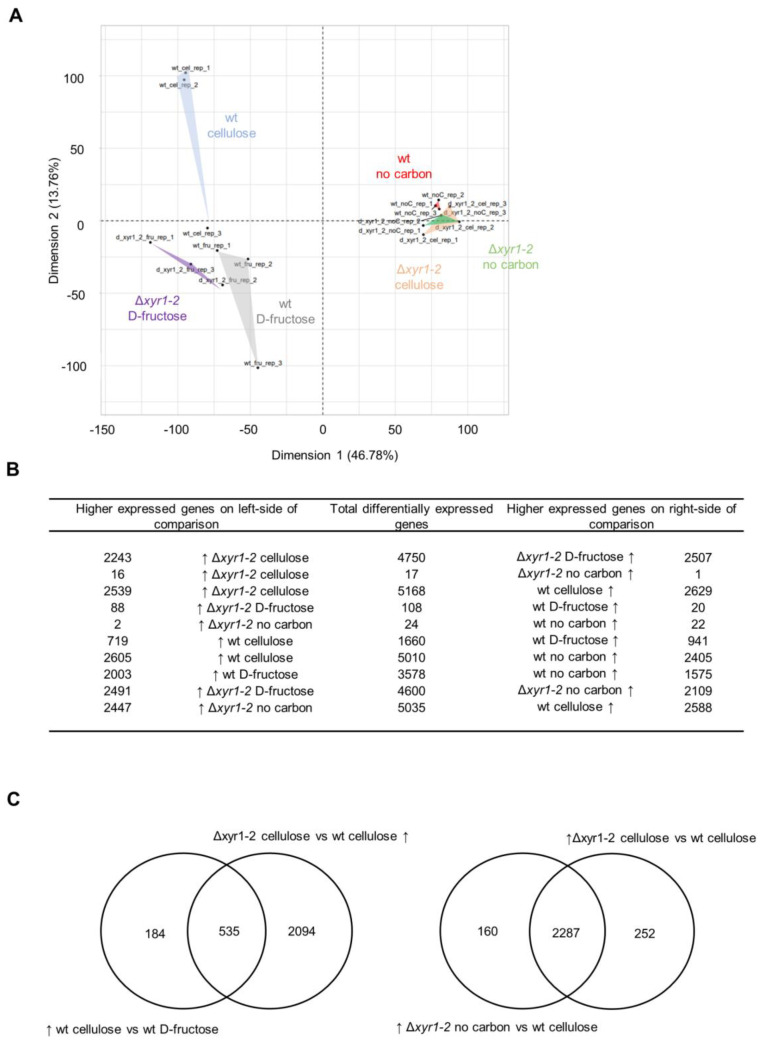
Overview RNA-Seq analysis of the *T. harzianum* Δ*xyr1* mutant. (**A**) Principal component analysis of the global transcriptomes of the *T. harzianum* wild type and Δ*xyr1-2* mutant 18 h after transfer to shake-flask cultures of D-fructose, no carbon, or cellulose. (**B**) Summary of the number of differentially expressed genes, and (**C**) Venn diagrams showing overlap in genes differentially expressed in particular comparisons. The arrow symbol ↑ indicates higher expressed genes in a particular condition in a comparison.

**Figure 5 biomolecules-14-00148-f005:**
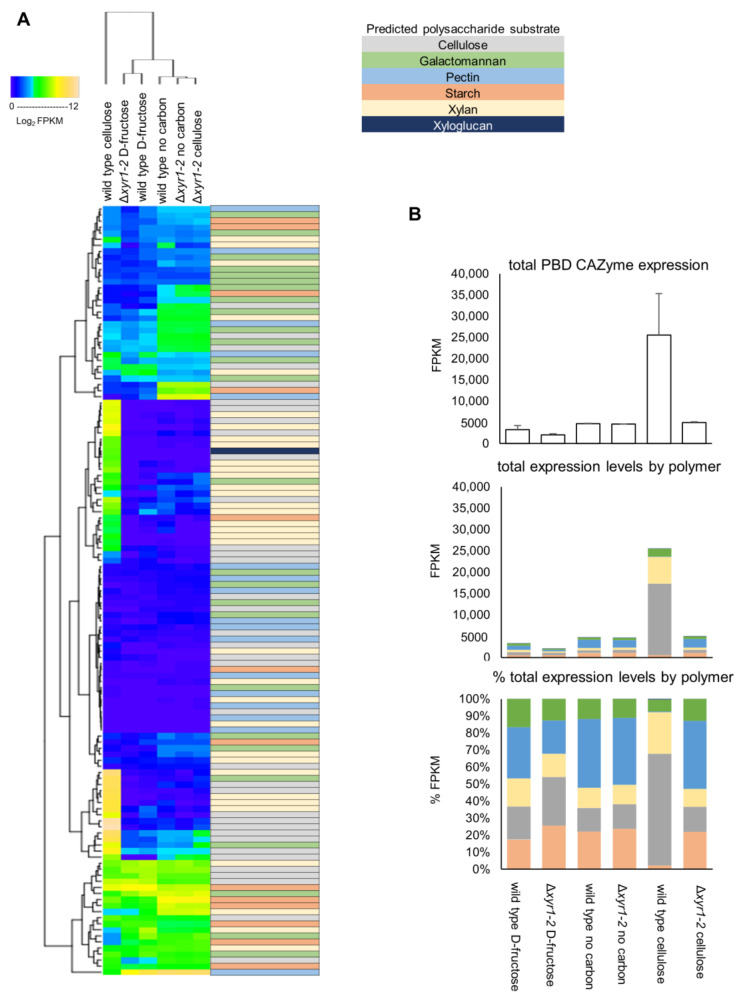
(**A**) Heatmap of the expression patterns of plant biomass-degrading (PBD) CAZy. The log2 FPKM values were clustered using hierarchical clustering. The predicted polysaccharide substrate that the PBD CAZyme acts on is color-coded. (**B**) Totals of PBD CAZy expression, and proportions attributed to activities towards particular polysaccharides present in plant biomass. Error bars on graphs represent standard errors (n = 3). See Appendix A for a high-resolution image of the heatmap where the gene IDs and annotations for each of the rows of the heatmap are visible.

## Data Availability

The RNA-Seq data from this study were deposited in the NCBI GEO database with the accession number GSE252008.

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
