# Peer review of "Growth, Enzymatic, and Transcriptomic Analysis of xyr1 Deletion Reveals a Major Regulator of Plant Biomass-Degrading Enzymes in Trichoderma harzianum"

_biomolecules, 2024, doi:10.3390/biom14020148_

Round 1
Reviewer 1 Report
Comments and Suggestions for Authors
In this paper, the authors discussed the quite a popular topic regarding enzymes plant biomass-degrading enzymes in Trichoderma harzianum. This manuscript requires minor revisions before it is ready for publication. Further detailed comments for consideration are provided below.
Comments:
Line 44: “Trichoderma” in italic;
Line 97: T. cf. guizhouense, it should be the full version, not the abbreviation;
Line 102: T. atroviride, it should be the full version, not the abbreviation;
Line 124: extra space;
Line 129, 279: “T. harzianum” in italic
Line 307, 361, : Trichoderma harzianum instead of T. harzianum
Line 476: The caption should be directly below the table.
Line 485: whole line should be in italic
Reviewer 2 Report
Comments and Suggestions for Authors
The article submitted by Wang et al. describes changes in biomass-degrading activities in a mutant strain of Trichoderma harzianum, in which the regulatory element xyr1 has been deleted. The article is clear and well-written. The presented results are convincing, and the drawn conclusions are solidly grounded. I consider it a significant contribution to the field of study.
Although other previous articles have described the results of overexpressing xyr1 in T. harzianum and the deletion of homologous genes as described herein in other systems, the work presented in this article allows for a more detailed understanding of the regulation of biomass degradation activities by this group of fungi. Therefore, I consider the results presented in this article relevant.
There are some considerations I would like to address to the authors for inclusion or revision in their manuscript:
1.- In the keywords section, it is generally advisable to avoid repeating words already in the title, as this does not contribute significantly to the subsequent computerized search for the article. I recommend not repeating 'Trichoderma harzianum' and 'XYR1.' Instead, I suggest exploring alternative keywords that can enhance the searchability of the article.
2.- There is some inconsistency in using italic and regular characters when citing species, genera, and genes. I urge the authors to review lines 36, 44, 79, 80, 81, 101-102, 279, 302, and 544.
3.- The authors describe the construction of the xyr1 deletion mutant in the Materials and Methods section; however, no data validating its construction are provided in the Results. I believe it is necessary to include a section at the beginning of the Results section describing the mutant and presenting evidence of its genetic structure. Some of the data presented in Figure S1 are more appropriate for the main text.
4.- The image of cellulose growth in Figure 1 is not very illustrative of what is described in the text in lines 280-281. Could it be improved?
5.- In Figure 2C, the authors investigate the proteins secreted in the presence of fructose as a carbon source in both the wild type and the mutant. What results would they expect using cellobiose as a carbon source? Can the authors comment on this result in the text?
6.- Lines 486-489. Including references to other articles in the Results section is not advisable. These sentences seem more appropriate for the Discussion section. I suggest the authors rephrase them and place them in the appropriate section.
7.- Line 551: This is the first time Fusarium is mentioned in the article. If so, please write the full genus name.
8.- Lines 611-612: The sentence is confusing. Please try to improve it.
In summary, I consider that the article is a significant contribution to gaining a clearer understanding of the regulation of biomass-degrading activities in Trichoderma. The points for discussion or correction are minor.
Reviewer 3 Report
Comments and Suggestions for Authors
Growth, enzymatic, and transcriptomic analysis of xyr1 deletion reveals a major regulator of plant biomass-degrading en- 3 zymes in Trichoderma harzianum
- great work by professionals! I have read the article several times and bow to the high level. There are no shortcomings in the work. Highest class!!!!!!!!!!!!!
Author Response
Response: We sincerely thank Reviewer #3 for his/her very positive comments.